

# Genetic and phenotypic differentiation of lumpfish (*Cyclopterus lumpus*) across the North Atlantic: implications for conservation and aquaculture

Benjamin Alexander Whittaker, Sofia Consuegra and Carlos Garcia de Leaniz

Department of Biosciences, Centre for Sustainable Aquatic Research, Swansea University, Swansea, Wales, United Kingdom

## ABSTRACT

Demand for lumpfish (*Cyclopterus lumpus*) has increased exponentially over the last decade, both for their roe, which is used as a caviar substitute, and increasingly also as cleaner fish to control sea lice in salmon farming. The species is classified as Near Threatened by the IUCN and there are growing concerns that over-exploitation of wild stocks and translocation of hatchery-reared lumpfish may compromise the genetic diversity of native populations. We carried out a comparative analysis of genetic and phenotypic variation across the species' range to estimate the level of genetic and phenotypic differentiation, and determined patterns of gene flow at spatial scales relevant to management. We found five genetically distinct groups located in the West Atlantic (USA and Canada), Mid Atlantic (Iceland), East Atlantic (Faroe Islands, Ireland, Scotland, Norway and Denmark), English Channel (England) and Baltic Sea (Sweden). Significant phenotypic differences were also found, with Baltic lumpfish growing more slowly, attaining a higher condition factor and maturing at a smaller size than North Atlantic lumpfish. Estimates of effective population size were consistently low across the North East Atlantic (Iceland, Faroe Islands and Norway), the area where most wild lumpfish are fished for their roe, and also for the aquaculture industry. Our study suggests that some lumpfish populations are very small and have low genetic diversity, which makes them particularly vulnerable to over-exploitation and genetic introgression. To protect them we advocate curtailing fishing effort, closing the breeding cycle of the species in captivity to reduce dependence on wild stocks, restricting the translocation of genetically distinct populations, and limiting the risk of farm escapes.

## INTRODUCTION

The control of parasitic sea-lice (*Lepeophtheirus salmonis*) is one of the most pressing problems facing salmon farming (*Torrissen et al., 2013*; *Treasurer, 2002*), as sea-lice have become resistant to chemical treatment (*Aaen et al., 2015*; *Lees et al., 2008*) and threaten the sustainability of the industry. Several species of cleaner fish have been used as an alternative to the use of antiparasitic therapeutants (*Treasurer, 2018*), but the lumpfish

Corresponding author
Carlos Garcia de Leaniz,
c.garciadeleaniz@swansea.ac.uk

(*Cyclopterus lumpus*) is proving the most useful as, in contrast to other cleaner fish like wrasse, it continues to feed on sea lice at low temperatures and is easier to rear in captivity (*Imsland et al., 2014*; *Powell et al., 2018a*). Demand for lumpfish has increased exponentially since 2012 (*Powell et al., 2018a*; *Treasurer, 2018*). However, nearly all lumpfish used in salmon farming are still derived from wild broodstock (*Jonassen et al., 2018a*), and as they are generally used in a single salmon production cycle (*Powell et al., 2018a*), satisfying aquaculture demands can put considerable pressure on wild stocks.

Lumpfish has been classified as Near Threatened in the IUCN Red List (*Lorance et al., 2015*), but information on the conservation status of different populations is very limited, and it is likely that some populations are already overexploited (*Myers & Sjare, 1995*; *Powell et al., 2018a*). Ripe females have traditionally been targeted for their roe, which is processed and sold as a cheap alternative to caviar, and while the Icelandic and Greenland lumpfish fisheries are closely monitored, others are largely unregulated (*Powell et al., 2018a*; *Kousoulaki, Treasurer & FitzGerald, 2018*). A strong reduction in catch per unit effort has been detected in some lumpfish fisheries over the last 25 years (*Lorance et al., 2015*), and there are concerns that removing additional spawners for the expanding lumpfish aquaculture industry could impact on some small populations (*Hedeholm, Post & Grønkjær, 2017*; *Powell et al., 2018a*; *Powell et al., 2018b*), as it has been reported for other cleaner fish fisheries (*Halvorsen et al., 2017*).

Stock movements represent an additional risk to wild lumpfish as large numbers of hatchery-reared lumpfish are being translocated across the North Atlantic to supply salmon farms (*Jonassen, Lein & Nytrø, 2018b*; *Treasurer et al., 2018*) and this could pose a potential threat to local populations. For example, over 85% of all lumpfish deployed in Scotland during 2017 originated from eggs imported from Iceland and Norway, and none came from local sources (*Treasurer et al., 2018*). In Ireland, 70% of lumpfish deployed during 2015–2016 were derived from eggs imported from Iceland and Norway (*Bolton-Warberg et al., 2018*), while in the Faroe Islands nearly all lumpfish used during 2014–2016 were of Icelandic origin (*Steinarsson & Arnason, 2018*; *Johannesen, Arge & Eliasen, 2018*; *Jonassen, Lein & Nytrø, 2018b*). There is a danger that if non-native lumpfish escape from salmon farms they could interbreed with local populations and result in genetic introgression (*Powell et al., 2018a*), as has been reported for farmed salmonids (e.g., *Consuegra et al., 2011*). Lumpfish translocations are likely to intensify in the near future (*Jónsdóttir et al., 2017*), and while escapes of lumpfish have not yet been reported, these seem largely inevitable in open salmon net-pens, as have already been documented for two species of wrasse (*Jansson et al., 2017*; *Faust et al., 2018*). Whether escapes have a genetic impact on local lumpfish populations will depend on the number of escapees, their reproductive success, and the extent of genetic differentiation between local and introduced fish, but none of these parameters are currently known.

Lumpfish are distributed across a vast marine area, extending to both sides of the North Atlantic and into the Baltic (*Davenport, 1985*; *Powell et al., 2018b*), and there is thus scope for substantial differentiation. Soon after hatching, the larvae attach to the substrate using a specialized suction cup, which probably limits larval dispersal (*Davenport, 1985*). Tagging studies suggest that, although adults can swim up to 49 km/day, some individuals

remain within a restricted 80 km range after +250 days at liberty (*Kennedy et al., 2015*). There is also evidence of homing (*Kennedy et al., 2015*), which will favour reproductive isolation and may result in stock differentiation. For example, spawning time may vary by two months within single populations (*Wittwer & Treasurer, 2018*), but as much as seven months among populations, from January in the English Channel (*Powell et al., 2018b*) to August near the Arctic circle (*Jónsdóttir, Imsland & Kennedy, 2018*). Population differences may also exist in growth and behaviour (*Johannesen, Arge & Eliasen, 2018*) and, as these are maintained under common rearing conditions (*Imsland et al., 2016*; *Bolton-Warberg et al., 2018*), they are likely to be inherited. Such differences suggest that lumpfish may form discrete populations, and that these may be adapted to local conditions. Yet, the extent of genetic differentiation in lumpfish is uncertain. Thus, while significant genetic differences have been found at large spatial scales using microsatellite markers (i.e., Canada vs Norway; (*Pampoulie et al., 2014*), populations at smaller scales appear to be relatively homogenous. For example, lumpfish sampled in the English Channel appear to be largely undifferentiated (*Consuegra et al., 2015*), as do fish sampled along the Norwegian coast (*Jónsdóttir et al., 2017*). In contrast, in Greenland two genetically distinct groups have been found in the north and south (*Garcia-Mayoral et al., 2016*), suggesting that there can also be some fine scale genetic structuring.

### Aims

There were three aims of this study. First, given their limited larval dispersal and evidence for homing, we hypothesised that lumpfish might display genetic isolation by distance, with populations closer together being more genetically similar than those further apart (*Rousset, 1997*). By sampling across the whole range, we aimed to estimate the level of genetic and phenotypic differentiation, and determine the patterns of gene flow across the species' range, at spatial scales relevant to management. Secondly, we wanted to know to what extent lumpfish translocations could pose a potential genetic risk to local populations. For this, we examined if Icelandic and Norwegian lumpfish, the two most common sources of lumpfish in aquaculture, were genetically distinct from lumpfish populations present in other salmon farming areas (i.e., Ireland, Canada, Scotland and the Faroe Islands) with which they might interbreed. As lumpfish aquaculture is very recent, our estimates of genetic structure of wild lumpfish populations might serve as genetic baselines against which the impact of farm escapees might be gauged, as has been done for Atlantic salmon (*Gilbey et al., 2017*). Finally, as some lumpfish populations may be endangered, we provide estimates of effective population size, and test for the existence of genetic bottlenecks to better understand their conservation status.

## MATERIAL AND METHODS

### Collection of samples

Fin tissue was obtained from 410 lumpfish originating from 15 sites across the species' range (Table 1) and were stored in 96% ethanol at −20 °C until analysis. Sites located within an 80 km radius (the estimated maximum range of dispersal *Kennedy et al., 2015*) were pooled together to minimise the risk of spatial pseudo-replication. Samples were
**Table 1  Details of study sites sampled for lumpfish.**

| Year | Country | Site | Lat. | Long. | N | Nb | | $N_A$ | $N_E$ | $N_{PA}$ | $H_O$ | $H_E$ | $F_{IS}$ |
|---|---|---|---|---|---|---|---|---|---|---|---|---|---|
| 2016 | USA | Frenchman Bay (FB) | 44.33 | −68.15 | 30 | – | mean | 6.0 | 3.100 | 0.300 | 0.566 | 0.613 | 0.100 |
| | | | | | | | ± SE | 0.775 | 0.548 | 0.213 | 0.058 | 0.044 | 0.050 |
| 2016 | USA | Cobscook Bay (CB) | 44.90 | −67.05 | 30 | – | mean | 6.1 | 3.452 | 0.100 | 0.640 | 0.668 | 0.078 |
| | | | | | | | ± SE | 0.862 | 0.456 | 0.100 | 0.063 | 0.038 | 0.055 |
| 2016 | Canada | Witless Bay (WB)* | 47.21 | −52.69 | 30 | 30 | mean | 6.7 | 3.459 | 0.400 | 0.630 | 0.673 | 0.080 |
| | | | | | | | ± SE | 0.870 | 0.425 | 0.163 | 0.049 | 0.036 | 0.050 |
| 2016 | Iceland | Hafnir (Ha) | 63.93 | −22.69 | 30 | – | mean | 5.5 | 2.971 | 0.000 | 0.637 | 0.643 | 0.019 |
| | | | | | | | ± SE | 0.500 | 0.222 | 0.000 | 0.041 | 0.031 | 0.050 |
| 2016 | Faroe Is. | Klasvík (Kl)* | 62.23 | −6.58 | 30 | – | mean | 6.8 | 3.713 | 0.200 | 0.668 | 0.700 | 0.065 |
| | | | | | | | ± SE | 0.359 | 0.453 | 0.133 | 0.049 | 0.030 | 0.047 |
| 2014 | Ireland | Ventry Bay (VB) | 52.20 | −10.12 | 30 | 26 | mean | 6.8 | 3.255 | 0.100 | 0.647 | 0.658 | 0.032 |
| | | | | | | | ± SE | 0.389 | 0.346 | 0.100 | 0.056 | 0.038 | 0.050 |
| 2017 | Scotland | Outer Hebrides (OH) | 58.16 | −6.38 | 30 | 18 | mean | 6.5 | 3.247 | 0.000 | 0.623 | 0.644 | 0.060 |
| | | | | | | | ± SE | 0.453 | 0.448 | 0.000 | 0.055 | 0.041 | 0.036 |
| 2015 | England | Weymouth (We) | 50.61 | −2.46 | 30 | 30 | mean | 5.8 | 2.979 | 0.000 | 0.607 | 0.597 | −0.012 |
| | | | | | | | ± SE | 0.593 | 0.423 | 0.000 | 0.062 | 0.059 | 0.053 |
| 2015 | England | Guernsey (Gu) | 49.47 | −2.59 | 30 | 30 | mean | 5.6 | 3.068 | 0.000 | 0.618 | 0.608 | 0.032 |
| | | | | | | | ± SE | 0.476 | 0.431 | 0.000 | 0.083 | 0.055 | 0.084 |
| 2017 | Norway | Namsen (Na) | 59.15 | 6.01 | 21 | 21 | mean | 6.3 | 3.080 | 0.100 | 0.576 | 0.614 | 0.105 |
| | | | | | | | ± SE | 0.539 | 0.470 | 0.100 | 0.076 | 0.050 | 0.078 |
| 2016 | Norway | Averøy (Av) | 63.05 | 7.48 | 30 | – | mean | 5.7 | 3.077 | 0.000 | 0.677 | 0.638 | −0.038 |
| | | | | | | | ± SE | 0.496 | 0.336 | 0.000 | 0.051 | 0.038 | 0.030 |
| 2015 | Norway | Rogaland (Ro) | 64.45 | 11.41 | 19 | – | mean | 4.5 | 2.625 | 0.000 | 0.600 | 0.594 | −0.056 |
| | | | | | | | ± SE | 0.342 | 0.291 | 0.000 | 0.071 | 0.026 | 0.046 |
| 2012 | Denmark | Køge Bay (KB)* | 55.46 | 12.18 | 30 | – | mean | 5.7 | 3.168 | 0.000 | 0.626 | 0.660 | 0.067 |
| | | | | | | | ± SE | 0.423 | 0.273 | 0.000 | 0.036 | 0.033 | 0.029 |
| 2017 | Sweden | Öland (Öl)* | 55.72 | 16.39 | 16 | 16 | mean | 4.7 | 2.838 | 0.100 | 0.548 | 0.592 | 0.110 |
| | | | | | | | ± SE | 0.423 | 0.320 | 0.100 | 0.077 | 0.061 | 0.073 |
| 2017 | Sweden | Gotska Sandön (GS)* | 57.95 | 18.97 | 24 | 24 | mean | 5.3 | 3.192 | 0.300 | 0.481 | 0.611 | 0.241 |
| | | | | | | | ± SE | 0.448 | 0.449 | 0.213 | 0.073 | 0.064 | 0.081 |

$N$, sample size for genetic analysis; $Nb$, sample size for biometric analysis; $N_A$, mean number of alleles (±SE); $N_E$, mean number of effective alleles; $N_{PA}$, number of private alleles; $H_O$, observed heterozygosity; $H_E$, expected heterozygosity; $F_{IS}$, fixation index.
*denotes deviation from HWE due to heterozygote deficiency after Bonferroni correction at $P < 0.0033$.

pooled from the Faroe Islands (Klasvík and Kollafjørður, c. 20 km), Denmark (Køge Bay and Mosede Havn, c. 13 km) and Sweden (survey hauls from Bornhölm to Öland, and from Gotland to Gotska Sandön). Pooled groups were named after the site contributing the largest number of samples. Biometric data on length (mm) and weight (g) were available for eight of the 15 sites (Table 1). Swansea University, College of Science Ethics Review Committee, provided full approval to Benjamin Whittaker for the genetic analysis of tissue samples already collected by other researchers (STU_BIOL_90920_181018132845_1).

## DNA extraction and amplification

DNA was extracted using the *Nexttec Isolation kit* (NextTec, Wellingborough, UK) following the manufacturer's protocol. The concentration of extracted DNA was quantified using a *Nanodrop 2000* (Thermo Fisher Scientific Inc., Waltam, MA, USA) and diluted with DNA free water to 50 ng/μl where necessary. A 2 μl of sample DNA was used
for amplification using a *QIAGEN Multiplex PCR kit* (QIAGEN, Manchester, UK) in a total reaction volume of 9 μl. Ten lumpfish specific microsatellite loci (*Clu*29, *Clu*34, *Clu*36, *Clu*45 and *Clu*12, *Clu*26, *Clu*33, *Clu*37, *Clu*40, *Clu*44 (*Skirnisdottir et al., 2013*) were genotyped in two separate multiplex reactions (Table S1). Amplification consisted of a single initial activation step at 95 °C for 15 min followed by eight cycles of touchdown PCR denaturation at 94 °C for 30 s, annealing from 64 °C or 60 °C to 56 °C in descending two-cycle steps of 2 °C and an extension at 72 °C for 90 s, 24 additional cycles with an annealing temperature of 56 °C and a single final extension at 60 °C for 30 min. An *Applied Biosystems ABI3130xl Genetic Analyser* (Applied Biosystems, UK) was used to resolve the fragments using *GeneScan* 500-LIZ(-250) as a size standard. Fragment length was established using *GeneMapper v5.0* (Applied Biosystems, Foster City, CA, USA). Genotyping consistency was validated by repeating PCR, fragment analysis and scoring for 10% of samples.

## Estimates of genetic diversity

We used *Microchecker v2.2.3* (*Van Oosterhout et al., 2004*) to identify null alleles, allele dropout and stutter peaks, and Bayescan v2.1 (*Foll & Gaggiotti, 2008*) to test for loci neutrality. *GENEPOP v4.2* (*Rousset, 2008*) was used to test for linkage disequilibrium, deviations from Hardy-Weinberg equilibrium, and to calculate allelic frequencies across populations. *GeneAlEx v6.502* (*Peakall & Smouse, 2012*) was used to assess the number of alleles ($N_A$), effective alleles ($N_E$), private alleles ($N_{PA}$), expected ($H_E$) and observed heterozygosity ($H_O$), and to carry out a Mantel test of genetic isolation by distance.

## Genetic differentiation, genetic structure and patterns of migration

We estimated the extent of genetic differentiation by calculating global and pairwise $F_{ST}$ values between populations by 1,000 permutations with a Bonferroni correction ($P < 0.00022$) using *Arlequin v3.5.2.2* (*Excoffier & Lischer, 2010*). We then conducted an Analysis of Molecular Variance (AMOVA) on pooled groups (separated more than 80 km) to partition genetic variation at three hierarchical levels (among populations, within populations, and among individuals), and estimated significance values using 1,000 permutations. To assess population structuring, a Bayesian cluster analysis was conducted in *STRUCTURE v2.3.4* (*Falush, Stephens & Pritchard, 2007*; *Hubisz et al., 2009*; *Pritchard, Stephens & Donnelly, 2000*) to estimate the most likely number of genetic clusters (K) informed by individual genotypes. Admixture models with *K* values ranging from 2 to 15 were run using twenty iterations, a burn-in length of 10,000 and 50,000 Markov Chain Monte Carlo repeat simulations to estimate the likelihood of each K value. Results were fed into *STRUCTURESELECTOR* (*Li & Liu, 2018*) to identify the most likely number of clusters present based on the median of means (MedMeaK), maximum of means (MaxMeaK), median of medians (MedMedK) and maximum of medians (MaxMedK) criteria (*Puechmaille, 2016*). In addition, we used a Bayesian cluster analysis implemented in *TESS v2.3.1* (*Chen et al., 2007*), which explicitly includes spatial information, to identify genetic discontinuities in our study area and estimates admixture proportions without assuming any predefined populations. For this, spatial coordinates were randomly
generated for each sample to fall within one standard deviation of the maximum and minimum latitude and longitude of each site (*Durand, Chen & François, 2008*). Admixture models were run with 50,000 total sweeps, 10,000 burn-in sweeps, and 200 runs per $K_{max}$ ranging from 2 to 15. The average Deviance Information Criterion (DIC) of the lowest 10 DIC values was calculated for each $K_{max}$ to assess the most likely number of clusters. Runs within 10% of the lowest (DIC) for a given $K_{max}$ were used for analysis. *CLUMMP v1.1.2* (*Jakobsson & Rosenberg, 2007*) was used to average variation between repeated iterations for the most likely $K$ values, and the resulting output was visualised using *DISTRUCT* v1.1.1 (*Rosenberg, 2004*). A neighbour joining tree was constructed with *Populations v1.2.32* (*Langella, 2002*) to further analyse population differentiation using Nei's standard genetic distance with 1,000 bootstraps per locus, and the resulting tree was visualised using *TreeView* (*Page, 2003*). Patterns of gene flow were estimated using *div-Migrate*, which defines a pool of simulated migrants based on the allele frequencies of each population pair and estimates asymmetric gene flow by measuring the genetic differentiation between each population and the migrants pool (*Sundqvist et al., 2016*). We used the *Nm* statistic (*Alcala, Goudet & Vuilleumier, 2014*) that combines elements of $G_{ST}$ and $D$ to calculate directional relative migration and assessed whether gene flow was significantly asymmetric between populations by running 5,000 bootstrap simulations.

### Effective population size and evidence of genetic bottlenecks
Estimates of effective population size ($N_e$) for sites containing at least 19 individuals were calculated using the Linkage Disequilibrium Model (LDM) with a critical value of 0.02 in *NeEstimator v2.1* (*Do et al., 2014*). Evidence of genetic bottlenecks was evaluated with *Bottleneck v2.1* (*Cornuet & Luikart, 1996*) using 1,000 replicates under the Two-Phase (TPM) and the Stepwise (SMM) Mutation Models to assess heterozygosity deficiency.

### Phenotypic variation
Variation in the Length–weight relationship between regions (West Atlantic, $n = 30$; East Atlantic, $n = 65$; English Channel, $n = 60$; Baltic Sea, $n = 40$), was examined by regression analysis on log-transformed data (*R Development Core Team, 2013*). We calculated relative weight (Wr) as the ratio of the observed weight divided by the predicted weight (from the regression obtained above) to obtain an index of body condition that is more appropriate for fish like lumpfish that have an unusual body shape (*Nahdi, Garcia de Leaniz & King, 2016*). The most plausible number of age classes represented in the samples, and the mean size at age (*Macdonald & Pitcher, 1979*) were calculated through mixture analysis of length-frequency data using PAST v3.17 (*Hammer, Harper & Ryan, 2001*). The Von Bertalanffy growth equation (*Kirkwood, 1983*) was fitted to estimate growth parameters in each region.

## RESULTS
### Population genetic diversity
All microsatellite loci were polymorphic. The mean number of alleles ($N_A$) ranged from 4.5 (Ro) to 6.8 (Kl, VB), mean expected heterozygosity ($H_E$) ranged from 0.592 (Öl) to 0.700
Table 2 Pairwise $F_{ST}$ values (lower) and Bonferroni adjusted $P$ values (upper; Bonferroni correction $P < 0.00022$) between 15 study populations of lumpfish distributed across the natural range of the species using 10 microsatellite loci.

| | FB | CB | WB | Ha | Kl | VB | OH | We | Gu | Na | Av | Ro | KB | Öl | GS |
|---|---|---|---|---|---|---|---|---|---|---|---|---|---|---|---|
| FB | | 0.036 | 0.000 | 0.000 | 0.000 | 0.000 | 0.000 | 0.000 | 0.000 | 0.000 | 0.000 | 0.000 | 0.000 | 0.000 | 0.000 |
| CB | 0.013 | | 0.000 | 0.000 | 0.000 | 0.000 | 0.000 | 0.000 | 0.000 | 0.000 | 0.000 | 0.000 | 0.000 | 0.000 | 0.000 |
| WB | 0.030 | 0.030 | | 0.000 | 0.000 | 0.000 | 0.000 | 0.000 | 0.000 | 0.000 | 0.000 | 0.000 | 0.000 | 0.000 | 0.000 |
| Ha | 0.130 | 0.112 | 0.117 | | 0.000 | 0.000 | 0.000 | 0.000 | 0.000 | 0.000 | 0.000 | 0.000 | 0.000 | 0.000 | 0.000 |
| Kl | 0.120 | 0.101 | 0.098 | 0.050 | | 0.081 | 0.000 | 0.000 | 0.000 | 0.000 | 0.000 | 0.000 | 0.000 | 0.000 | 0.000 |
| VB | 0.117 | 0.093 | 0.102 | 0.042 | 0.011 | | 0.000 | 0.000 | 0.000 | 0.018 | 0.000 | 0.000 | 0.243 | 0.000 | 0.000 |
| OH | 0.152 | 0.111 | 0.124 | 0.049 | 0.034 | 0.021 | | 0.000 | 0.009 | 0.252 | 0.000 | 0.000 | 0.000 | 0.000 | 0.000 |
| We | 0.177 | 0.154 | 0.146 | 0.065 | 0.056 | 0.042 | 0.024 | | 0.324 | 0.000 | 0.000 | 0.000 | 0.000 | 0.000 | 0.000 |
| Gu | 0.188 | 0.160 | 0.152 | 0.083 | 0.060 | 0.057 | 0.014 | 0.003 | | 0.009 | 0.000 | 0.000 | 0.000 | 0.000 | 0.000 |
| Na | 0.157 | 0.121 | 0.128 | 0.080 | 0.035 | 0.020 | 0.004 | 0.029 | 0.029 | | 0.036 | 0.000 | 0.018 | 0.000 | 0.000 |
| Av | 0.153 | 0.122 | 0.108 | 0.102 | 0.027 | 0.021 | 0.042 | 0.061 | 0.061 | 0.018 | | 0.000 | 0.000 | 0.000 | 0.000 |
| Ro | 0.142 | 0.132 | 0.138 | 0.057 | 0.039 | 0.041 | 0.043 | 0.065 | 0.061 | 0.048 | 0.075 | | 0.000 | 0.000 | 0.000 |
| KB | 0.113 | 0.085 | 0.095 | 0.034 | 0.021 | 0.004 | 0.028 | 0.048 | 0.067 | 0.022 | 0.037 | 0.046 | | 0.000 | 0.000 |
| Öl | 0.194 | 0.139 | 0.176 | 0.087 | 0.129 | 0.115 | 0.097 | 0.105 | 0.113 | 0.126 | 0.152 | 0.149 | 0.088 | | 0.216 |
| GS | 0.187 | 0.154 | 0.181 | 0.097 | 0.140 | 0.132 | 0.134 | 0.136 | 0.152 | 0.159 | 0.175 | 0.152 | 0.110 | 0.011 | |

(Kl), and mean $F_{IS}$ varied from $-0.056$ (Ro) to 0.110 (Öl) across all loci (Table 1). Initial analysis suggested that null alleles might be present at multiple loci (*Clu34, Clu36, Clu12, Clu33, Clu37* and *Clu40,* Table S2 ). However, repeatedly removing each locus in turn showed little variation in $F_{ST}$ values (Table S3–S8), and therefore all markers were retained for further analyses. No evidence of departures from neutrality or linkage disequilibrium was found after Bonferroni corrections for multiple tests (*Rice, 1989*). Deviations from Hardy-Weinberg equilibrium (HWE) were detected at five of the 15 sites (Table 1), but these involved only 12% of loci after Bonferroni correction (Table S9). The mean number of private alleles ($N_{PA}$) was relatively low, ranging from 0.00 to 0.40, with sites in the West Atlantic (FB = 0.30, WB = 0.40) and Baltic Sea (GS = 0.30) showing the highest values.

## Population structure and gene flow

Global $F_{ST}$ was 0.095 ($P < 0.001$) indicating a moderate but significant degree of genetic differentiation for a marine fish (*Hartl, Clark & Clark, 1997*; *Ward, 2000*; *Hutchinson, Carvalho & Rogers, 2001*). Results of AMOVA indicated that 83.5% of molecular variation was due to variation within individuals, 7% amongst individuals within populations, and 9.5% amongst populations (Table S10). Pairwise $F_{ST}$ showed a significant level of genetic differentiation across most populations (Table 2), but populations closer together were genetically more similar after Bonferroni correction. On the basis of $F_{ST}$ values, the strongest differentiation was found between West Atlantic and Baltic Sea populations. Results of a Mantel test support the existence of a significant, albeit weak, isolation by distance ($R^2 = 0.1229$, $P = 0.01$).

The most likely number of genetically distinct groups ($K$) ranged from $K = 5$ (MedMedK, MedMeaK) to $K = 6$ (MaxMedK, MaxMeaK) using *STRUCTURESELECTOR* (Fig. S1). Spatial cluster analysis using TESS suggested a $K_{max} = 10$ (Fig. S1), though only
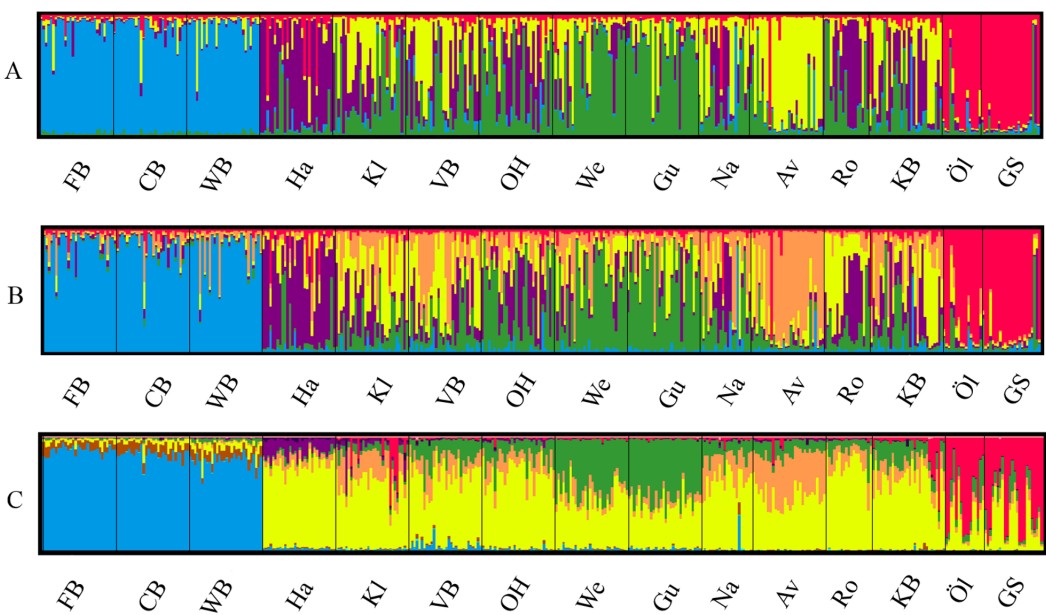

**Figure 1 Lumpfish genetic structure.** Lumpfish genetic structuring according to (A) STRUCTURES-ELECTOR with MedMedK and MedMeanK, $K = 5$, (B) STRUCTURESELECTOR with MaxMedK and MaxMeanK, $K = 6$, and (C) TESS with $K_{max} = 10$ based on lowest mean DIC value. Each bar represents one individual with colours indicating probability of belonging to different genetically distinct groups.

six of these genetic groups showed substantial representation, and four groups contributed only 3.3% to the genetic background. Distinct clusters were detected in the West Atlantic and Baltic Sea by both *STRUCTURE* and *TESS*, with a greater level of admixture across the East Atlantic (Figs. 1A–1C). Results were consistent in attributing a genetically unique pattern to the Mid Atlantic, English Channel clusters and a Norwegian site at Averøy. A neighbour joining tree (Fig. 2) showed similar patterns to that of the structuring analyses, highlighting the separation between the West Atlantic and Baltic Sea populations, and the higher degree of admixture within the East Atlantic group.

The effective number of migrants ($N_m$) ranged from 1.00 between sites in the English Channel to 0.03 between sites in the West Atlantic and Baltic Sea. The exchange of migrants was much higher within genetic clusters than among clusters (Table S11), with the highest levels of gene flow found within the East Atlantic and within the English Channel (Fig. 3). The only evidence of moderate asymmetric gene flow was from Norway towards the Faroe Islands ($N_m = 0.507$, $P < 0.05$).

## Effective population size and evidence of genetic bottlenecks

Estimates of effective population size ($N_e$) based on a Linkage Disequilibrium Model (LDM) varied from 19 (Norway) to 70,148 (Denmark; Table S12). Sites with low $N_e$ values (<75) were found across Iceland, Faroe Islands and Norway (Fig. 3). A significant deficiency of heterozygotes was identified in Ireland and Scotland using the Single Mutation Model (SMM) in *Bottleneck* (Wilcoxon signed-rank test, $P = 0.0033$ after Bonferroni correction),

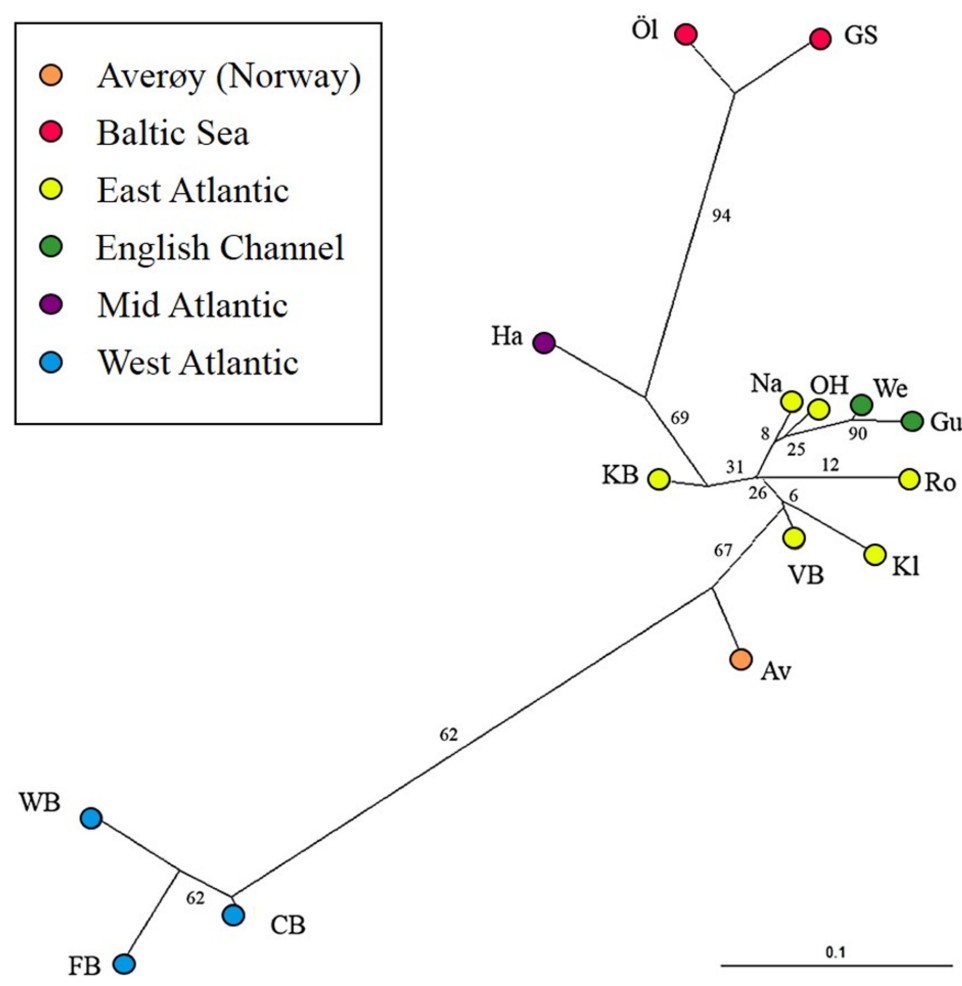

**Figure 2 Neighbour joining tree.** Neighbour joining tree (based on Nei's Standard Genetic Distance) of 15 lumpfish populations genotyped with 10 microsatellite loci. Branch values denote % support from 1,000 bootstraps.

suggesting that these populations could have undergone a recent genetic bottleneck (Table S13), but this was not detected by the Two-Phase Model of Mutation (TPM).

## Phenotypic variation

The relationship between length and weight differed significantly between regions ($F_{4,192} = 917.2$, $P < 0.001$; Fig. 4). Lumpfish in the Baltic Sea were heavier relative to their size than lumpfish in the East Atlantic and the English Channel (pairwise comparisons: Baltic—East Atlantic, estimate $= -0.090 \pm 0.036$, $t = -2.530$, $P = 0.012$; Baltic—English Channel, estimate $= -0.145 \pm 0.046$, $t = -3.171$, $P = 0.002$), but were similar to those in the West Atlantic (pairwise comparison Baltic—West Atlantic, estimate $= -0.094 \pm 0.050$, $t = -1.891$, $P = 0.060$). The relative weight of lumpfish differed

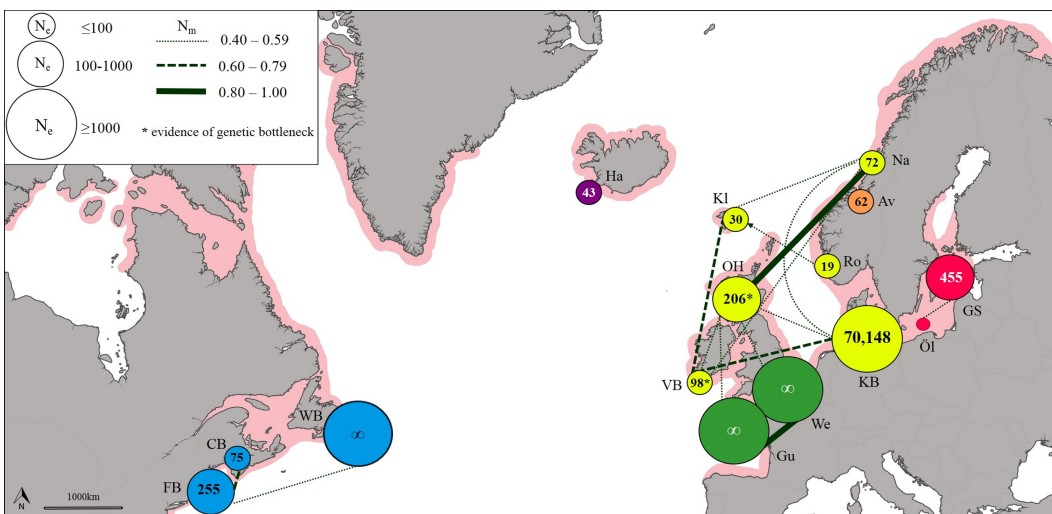

**Figure 3** Patterns of gene flow among lumpfish populations with colours indicating genetic groups, symbol size proportional to effective population size, line thickness proportional to effective number of migrants, and shaded area representing spawning distribution.

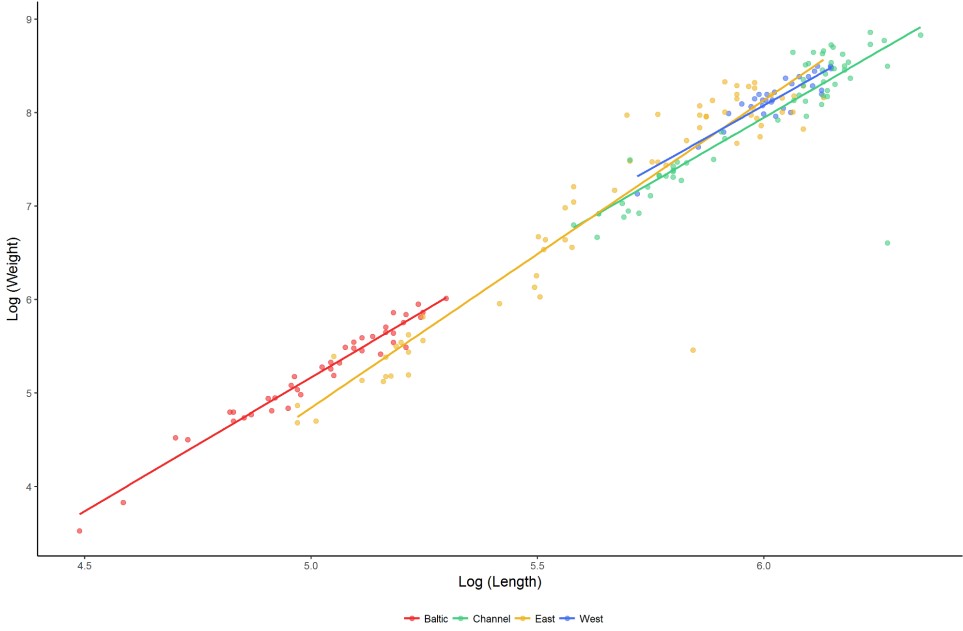

**Figure 4** Length–weight relationships ($\log_{10}$ scale) for lumpfish sampled in the Baltic Sea, English Channel, East Atlantic and West Atlantic.

**Table 3 Von Bertalanffy growth parameters.** ($L\infty$, asymptotic length; $t_0$, initial condition parameter, and $K$, Brody growth rate or curvature parameter) and estimated mean weight at first maturity ($\pm$ 95 CI) for lumpfish from different genetically distinct regions).

| Region | Von Bertalanffy Growth parameters. | | | Weight at 1st maturity (g) |
|---|---|---|---|---|
| | $L_\infty$ (mm) | $t_0$ | $K$ (yr$^{-1}$) | |
| Baltic Sea | $200 \pm 6$ | $0.14 \pm 0.02$ | $0.51 \pm 0.02$ | $150 \pm 12.5$ |
| East Atlantic | $461 \pm 14$ | $0.36 \pm 0.23$ | $0.56 \pm 0.09$ | $2,019 \pm 265.5$ |
| English Channel | $571 \pm 22$ | $-1.08 \pm 0.59$ | $0.35 \pm 0.20$ | $3,007 \pm 519.5$ |

between regions ($F_{3,191} = 2.841$, $P = 0.039$) and was highest in the Baltic Sea and the West Atlantic, and lowest in the East Atlantic and the English Channel.

Mixture analysis identified multiple plausible age classes present amongst lumpfish sampled in the Baltic Sea (seven age classes), East Atlantic (four age classes) and English Channel (three age classes), but only a single plausible age class in the West Atlantic. Based on the parameters of the Von Bertalanffy Growth equation, the maximum age was estimated to be 6.0 years for Baltic populations, 5.7 yrs for populations in the East Atlantic and 7.5 yrs for southern populations spawning in the English Channel. Fitted growth equations differed significantly between regions (Table 3), with lumpfish in the Baltic Sea showing the slowest growth and those in the English Channel showing the fastest.

## DISCUSSION

Our study reveals a significant degree of population structuring in lumpfish, which is consistent with moderate isolation by distance—as seen in other marine fish (e.g., *Ward, 2000*; *Hutchinson, Carvalho & Rogers, 2001*), and which should inform the translocation of this species across salmon farms. Genetically distinct groups were found in the West Atlantic (USA, Canada), Mid Atlantic (Iceland), East Atlantic (Faroe Islands, Ireland, Scotland, Norway, Denmark), English Channel, Averøy (Norway) and Baltic Sea (Sweden). Whilst significant gene flow was detected within each of these groups, little exchange of migrants was found between these areas.

Our results also indicate the existence of significant phenotypic differences across the range, that mimic to some extent the observed genetic differences. Lumpfish typically mature at 3–5 years of age, although some populations mature after only 2 years and some individuals may exceptionally live up to 14 years (reviewed in *Powell et al., 2018a*; *Powell et al., 2018b*). Lumpfish from the Baltic Sea were not only genetically distinct, they were also smaller, grew at a slower rate, and weighed more relative to their size than lumpfish from the North Atlantic. Although our growth estimates were based on length frequency data and did not distinguish between males and females, they are in line with estimates based on mark and recapture studies in Norway and Iceland ($L_\infty = 527 \pm 64$ mm, $K = 0.26 \pm 0.14$ year$^{-1}$; *Kasper et al., 2014*), and suggest that Baltic lumpfish grow more slowly and mature at a much smaller size (c. 150 g) than lumpfish from the North Atlantic (2.0–3.0 kg). The slow growth shown by Baltic lumpfish may be of interest for selective breeding programmes in aquaculture, as slow growing cleaner fish may be better suited for feeding on sea lice

(*Powell et al., 2018a*); however, given their markedly different genetic structure, care should be taken to ensure that Baltic lumpfish do not introgress into Atlantic lumpfish populations via farm escapes.

*Pampoulie et al. (2014)* first suggested that lumpfish in the West and East Atlantic were separated by cold southward polar currents, and that populations in the Baltic Sea may have become isolated during the Last Glacial Maximum. Though our analysis supports this broad division, it also indicates a finer population structure, revealing that lumpfish in the Mid Atlantic and English Channel are genetically distinct from other populations. The conclusion of our genetic analyses is consistent with recent tagging studies in Norway and Iceland showing that whilst lumpfish can move offshore to feed, they return to spawn in their home waters (*Kennedy et al., 2015*; *Kennedy et al., 2016*) and do not migrate between Iceland and Norway (*Kasper et al., 2014*). There is little information on southern lumpfish populations, though lumpfish in the English Channel appear to spawn earlier in the season than populations further north (*Powell et al., 2018b*), probably due the warmer temperatures and better feeding opportunities, which are known to influence maturation and spawning of lumpfish (*Hedeholm, Post & Grønkjær, 2017*). It is thus possible that the warmer waters found at the species' southern range may favour an early spawning and lead to some degree of reproductive isolation, hence limiting gene flow along a latitudinal gradient. With the exception of the Averøy population, the remaining sites in the East Atlantic appear to be genetically uniform, as reported along the Norwegian coast (*Jónsdóttir et al., 2017*).

The level of genetic diversity, and therefore the ability to adapt and respond to selection, differed substantially among regions. Our estimates of effective population size, the first for this species, were particularly low across the North East Atlantic (Iceland, $N_e = 43$; Faroe Islands, $N_e = 30$; Norway, mean $N_e = 51$), and some evidence of genetic bottlenecks was also detected at sites in Ireland and Scotland, though the evidence for this was not strong. The North East Atlantic supports one of the largest lumpfish roe fisheries (*Jónsdóttir, Imsland & Kennedy, 2018*), with a production of 4,000 tonnes of roe per year (*Johannesson, 2006*). Given a maximum yield of c. 4 kg roe/female (*Johannesson, 2006*), this level of harvest likely surpasses 1 million mature females every year. Harvesting for lumpfish roe is both size and sex-selective, which increases the vulnerability of populations to over-exploitation (*Hoenig & Hewitt, 2005*; *Ratner & Lande, 2001*) and may explain the low estimates of effective population size found across this area. The genetic diversity of North East Atlantic populations appear to be small and reducing pressure on these stocks would decrease the risk of over exploitation.

Although our estimates of effective population size are based on a relatively low number of genetic markers (selected to allow comparisons with previous studies, *Pampoulie et al., 2014*; *Jónsdóttir et al., 2017*), and need to be interpreted with caution, estimates based on the LD method using similar conditions to ours (10 loci with an average of 10 alleles and a sample size of 50 individuals) have proved accurate for small effective population sizes ($N_e \sim 100$) *Waples & Do, 2010*). This was the case for most of our populations, with the exception of three $N_e$ estimates that resulted in infinity. This is likely to have been due to the use of the LD method, which is generally efficient for small populations, but may

be unable to detect weak drift signals in larger populations, and can be very sensitive to small violations of model assumptions (*Waples & Do, 2010*). We therefore recommend that further analyses are carried out with a larger number of markers (e.g., genome wide distributed SNPs), as this may allow more accurate estimates of effective population sizes and other demographic parameters.

## Conclusions and management implications

By 2020 c. 50 million lumpfish will be required by the salmon farming industry (*Powell et al., 2018a*; *Treasurer, 2018*) and most of these will come from the stripping of wild broodstock (*Wittwer & Treasurer, 2018*) caught in Iceland and Norway, and then shipped as eggs or larvae to salmon farms elsewhere. Information on lumpfish escapees is lacking but corkwing wrasse (*Symphodus melops*) deployed as cleaner fish in Norway have recently been found to escape and hybridise with local populations (*Faust et al., 2018*), and the same could happen with lumpfish. Efforts should thus be made to reduce the risk of lumpfish escaping from fish farms and interbreeding with local populations, as high propagule pressure associated with open-net pens is the single most important factor determining the impact of escapees (*Consuegra et al., 2011*).

Our study suggests that lumpfish translocations should be restricted within genetically homogenous groups to reduce the risk of genetic introgression between native and non-native populations. In this sense, lumpfish from some areas of Norway, and particularly from Iceland, may be ill-suited for deployment in Ireland, Scotland and the Faroe Islands, and vice-versa. Ultimately, closing the breeding cycle of the species in captivity, and producing sterile lumpfish for deployment in salmon farms, must be a research priority for both the conservation of the species and the cleaner fish industry (*Powell et al., 2018a*), as this will lessen dependency on wild broodstock and reduce the risk of genetic introgression.

## ACKNOWLEDGEMENTS

We are indebted to Majbritt Bolton-Warberg, Danny Boyce, Werner Forster, Gus Galloway, Ása Johannesen, Lars Jørgen Ulvan, Niklas Larson, Marine Harvest Scotland, Michael Pietrak, Adam Rainsden and Peter Rask for supplying tissue samples. We are also grateful to Chloe Robinson, Niall Coates, Christine Gray, Craig Pooley and Ian Tew for assistance in optimising primers and processing samples.

### Funding

This work was funded by Marine Harvest Scotland through the LUMPFISH project and the Welsh Government via the European Regional Development Fund (SMARTAQUA Operation). There was no additional external funding received for this study. The funders had no role in study design, data collection and analysis, decision to publish, or preparation of the manuscript.

## Grant Disclosures

The following grant information was disclosed by the authors:

Marine Harvest Scotland.

European Regional Development Fund (SMARTAQUA Operation).

## Competing Interests

The authors declare there are no competing interests.

## Author Contributions

- Benjamin Alexander Whittaker performed the experiments, analyzed the data, prepared figures and/or tables, authored or reviewed drafts of the paper, approved the final draft.
- Sofia Consuegra conceived and designed the experiments, analyzed the data, contributed reagents/materials/analysis tools, authored or reviewed drafts of the paper, approved the final draft, wrote the grant and secured funding.
- Carlos Garcia de Leaniz conceived and designed the experiments, analyzed the data, contributed reagents/materials/analysis tools, prepared figures and/or tables, authored or reviewed drafts of the paper, approved the final draft, wrote the grant and secured funding.

## Animal Ethics

The following information was supplied relating to ethical approvals (i.e., approving body and any reference numbers):

Swansea University College of Science Ethics Review Committee provided full approval to Benjamin Whittaker for the genetic analysis of lumpfish tissue samples (STU_BIOL_90920_181018132845_1).

## Data Availability

Whittaker, Ben; Consuegra, Sofia; Garcia de Leaniz, Carlos (2018): Lumpfish Genotype Data.xlsx. figshare. Dataset. https://doi.org/10.6084/m9.figshare.7000355.v2.

## Supplemental Information

Supplemental information for this article can be found online at http://dx.doi.org/10.7717/peerj.5974#supplemental-information.

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
