# Peer review of "Genetic and phenotypic differentiation of lumpfish (Cyclopterus lumpus) across the North Atlantic: implications for conservation and aquaculture"

_PeerJ, doi:10.7717/peerj.5974_

## Round 0.1 · original submission · Minor Revisions

We have now received two reviews for your manuscript. Both reviewers felt that your manuscript deserves to be published once the modifications suggested are incorporated. In particular, as you can assess from their comments, they would like to see the details of the methods better outlined, including a clearer link with the study objectives. More specifically, the first reviewer questioned the relatively low number of loci used and asked that the revised discussion includes a section concerning this issue and its impact on the results (effective size). The second reviewer specifically asked for a clarification of one of the research questions and had concerns regarding Bayesian clustering analyses that need to be clearly addressed in your revision. Revised figures should also be provided to better illustrate the data.

Reviewer 1 ·

Basic reporting

In general, the manuscript was well written.

Experimental design

The methods section lacks sufficient detail. The relevance of some of the statistical tests to the overall objectives of the study should be made clearer.

Validity of the findings

The number of loci used is quite low for robust estimation of demographic parameters. This needs to be addressed in the discussion.

Additional comments

Minor Comments
Line 102. Which classes of markers have been used in these earlier studies?
Line 110. The first sentence should read, “There were three basic aims of this study.”
Line 116, 121. New paragraphs are not needed here.
Line 119. Describe what is meant by “genetic baseline”.
Line 129. A map should be included here to show lumpfish distribution and sampled sites.
Line 128. Clarify how were individuals assigned to populations for AMOVA analysis?
Line 144. Were this loci designed for use in lumpfish? This should be stated here.
Line 189. What does alpha stand for here? It is unclear what this matrix is being used for. In general more explanation is needed regarding how divMigrate estimates Nm.
Line 193. How well does the LDM method estimate Ne based on only 10 loci and small sample sizes?
Line 197. Why test for population expansion here? Is this a test for bottlenecks? This needs more explanation.
Line 203. What is the life-span and age of reproduction for lumpfish? Did you examine sex ratios?
Line 240. This FST seems very low. What is considered significant differentiation? A reference should be used to support this.
Line 243. The AMOVA results should be shown in a table.
Line 255. This should be a “neighbor joining tree”.
Line 299. Many of the FST values were very low.
Line 309. Remove “did”.
Figure 2. Specify in the legend what the numbers are on the branches. Should the legend read Neighbor joining tree?
Figure 3. Populations should be labelled here to be consistent with Figure 2. One of the populations has an effective population size of infinity. This usually means that the data was uninformative. This should be addressed in the discussion.

Reviewer 2 ·

Basic reporting

Overall this manuscript is well presented, clearly written and concise. The figures and tables are relevant and the raw data are supplied as supplementary material.

Experimental design

The research questions are well defined except the third one (lines 122-125) where the authors propose to estimate ‘effective population size and test for genetic bottlenecks to better understand conservation status and the extent to which gene flow could mitigate the impact of over -exploitation’. The latter statement is unclear as the focus of the ms is rather on the preservation of distinct genetic units with potential local adaptation (putatively linked to phenotypic differentiation) hence gene flow (artificial by translocation?) would not seem the best solution to ‘mitigate the impact of overexploitation’ and fishery regulations would seem more appropriate?

The statistical analyses are relevant and well performed, and methods are mostly described with sufficient detail.

I have only some concerns with the Bayesian clustering analyses of population structure. The authors used the programs Structure and Tess to identify the number of clusters (K) in their data set. However, with Structure they used the new estimators developed by Puechmaille (2016) to infer K even though those estimators require the a priori information of geographic origin of individuals (or the membership to a given population) to be calculated. Hence this analysis can be seen as partially redundant with the one performed with Tess that explicitly uses spatial information of the localization of individuals to infer K. An alternative would be to analyze Structure output with no a priori on the geographic origin of individuals by using the direct posterior probability for each K or the deltaK estimator. In addition, it is not clear how the spatial information was incorporated in the Tess analysis as it seems that the precise location where each individual fish was collected was not known as only one location is indicated for each site even though those sites may cover large areas.

Validity of the findings

The conclusions are well stated and limited to supported results.

Additional comments

I believe this manuscript deserves to be published once the modifications suggested above are incorporated.

---

## Round 0.2 · accepted · Accept

The authors are commended for their revisions on the manuscript. I am pleased to recommend it for publication.

#